# Endoplasmic Reticulum Stress Signaling and the Pathogenesis of Hepatocarcinoma

**DOI:** 10.3390/ijms22041799

**Published:** 2021-02-11

**Authors:** Juncheng Wei, Deyu Fang

**Affiliations:** Department of Pathology, Northwestern University Feinberg School of Medicine, Chicago, IL 60611, USA

**Keywords:** hepatocellular carcinoma, endoplasmic reticulum stress, unfolded protein response

## Abstract

Hepatocellular carcinoma (HCC), also known as hepatoma, is a primary malignancy of the liver and the third leading cause of cancer mortality globally. Although much attention has focused on HCC, its pathogenesis remains largely obscure. The endoplasmic reticulum (ER) is a cellular organelle important for regulating protein synthesis, folding, modification and trafficking, and lipid metabolism. ER stress occurs when ER homeostasis is disturbed by numerous environmental, physiological, and pathological challenges. In response to ER stress due to misfolded/unfolded protein accumulation, unfolded protein response (UPR) is activated to maintain ER function for cell survival or, in cases of excessively severe ER stress, initiation of apoptosis. The liver is especially susceptible to ER stress given its protein synthesis and detoxification functions. Experimental data suggest that ER stress and unfolded protein response are involved in HCC development, aggressiveness and response to treatment. Herein, we highlight recent findings and provide an overview of the evidence linking ER stress to the pathogenesis of HCC.

## 1. Introduction

The endoplasmic reticulum (ER) is a multifunctional organelle within which protein folding, lipid biosynthesis, and calcium storage occur [1,2,3,4]. The ER in hepatocytes has a remarkable capacity to adapt to extracellular and intracellular challenges [5,6,7]. However, in humans, numerous disturbances impair hepatocytes’ ability to handle protein folding and proteostasis, which disrupts ER homeostasis and leads to misfolded protein accumulation, ER stress and unfolded protein response (UPR) [8,9,10,11,12]. ER stress and subsequent adaptive UPR activation are important in the pathogenesis of chronic liver disease, ranging from steatosis and nonalcoholic fatty liver disease (NAFLD) to hepatocellular carcinoma (HCC) [13,14,15,16,17]. The key interaction between the UPR and metabolic liver disease has been comprehensively discussed in previous reviews [18,19,20]. This review compiles the recent findings supporting the functional impact of ER stress signaling on the pathogenesis of HCC.

## 2. ER Stress and UPR

The UPR plays a critical role in mediating cellular adaptation to ER stress [21,22]. UPR acts in concert to increase ER content, expands the ER protein folding capacity, degrades misfolded proteins and reduces the load of new proteins to the ER [23,24,25,26]. Activation of the UPR depends on the ER transmembrane proteins and sensors including inositol-requiring enzyme 1 (IRE1α), protein kinase R (PKR)-like endoplasmic reticulum kinase (PERK), activating transcription factor 6 (ATF6) and cyclic adenosine monophosphate (cAMP)-responsive element-binding protein H (CREBH) [27,28,29,30] (Figure 1). Both PERK and IRE1α are type I transmembrane proteins with similar ER luminal domain structures and a cytosolic Serine/threonine kinase domain, whereas ATF6α is a type II transmembrane protein that contains a cytosolic cyclic AMP response element-binding protein (CREB)–ATF basic leucine zipper domain [31,32,33,34]. IRE1α, PERK, ATF6 and CREBH are ultimately responsible for the activation of a set of transcription factors (TF), including spliced X-box binding protein 1 (XBP1s), activating transcription factor 4 (ATF4), CCAAT enhancer-binding protein (C/EBP) homologous protein (CHOP), nuclear factor κB (NF-κB) and activator protein 1 (AP-1), through a complicated and nonparallel process [35,36,37,38,39].

Hepatocytes, like other secretory cells, have a well-developed ER [18,40]. Due to their high protein and lipid synthesis, hepatic UPR is activated by rhythmic or transient physiological conditions (e.g., feeding-fasting cycles) to regulate lipid and glucose homeostasis [5,41,42,43,44]. However, the UPR is also chronically activated by irreversible or chronic stress (e.g., viral infection and obesity) and causes hepatic dysfunction, leading to the pathogenesis of the HCC, including viral hepatitis, alcoholic fatty liver disease, non-alcoholic fatty liver disease (NAFLD) and alpha1-antitrypsin deficiency [13,14,15,16,17].

### 2.1. Serine/Threonine-Protein Kinase/Endoribonuclease IRE1α and Transcription Factor Xbp1s

IRE1 is the most conserved ER stress sensor, with two isoforms identified in humans: IRE1α and IRE1β [28]. IRE1α is a ubiquitously expressed and dual function transmembrane protein with serine/threonine kinase and endoribonuclease (RNase) activities on its cytosolic tail [28,45]. Upon binding with ER chaperone Hsp47 and dissociation from BiP, IRE1α oligomerizes, trans-autophosphorylates and activates its RNase activity [46,47]. The mRNA encoding XBP1 undergoes a cytosolic unconventional splicing reaction by IRE1α and the tRNA ligase RTCB at a stem-loop structure that results in the translation of an active transcription factor, termed spliced XBP1 (XBP1s) [48]. Transcriptional targets of XBP1s include genes that encode functions in ER protein folding, secretion, ER-associated degradation (ERAD), ER biogenesis and lipid synthesis [28,41,49].

In mice, genetic ablation of either IRE1α or XBP1 results in embryonic lethality with major liver defects, which indicate that IRE1α/Xbp1s pathway is involved in hepatic physiological functions other than its involvement in ER stress response [50,51]. XBP1 deficiency leads to diminished hepatocyte development and prominent apoptosis [51]. One of the specific target genes of XBP1s in the liver is identified as αFP, which may be a regulator of hepatocyte growth [51]. IRE1α knockout results in a reduction in vascular endothelial growth factor and fetal liver hypoplasia49. Liver conditional knockout studies have helped to understand the important role of IRE1α/Xbp1s pathway in liver physiology [52]. XBP1 deficiency in the liver results in a profound compromise of de novo hepatic lipid synthesis, thereby leading to decreased serum triglycerides (TG), cholesterol and free fatty acids (FFAs) without causing hepatic steatosis [52]. As a transcription factor, XBP1s directly regulates the expression of a subset of lipogenic genes including stearoyl-CoA desaturase 1 (SCD1), Acetyl-CoA carboxylase 2(ACC2) and Diacylglycerol O-acyltransferase 2 (DGAT2) [52]. Moreover, IRE1α-mediated mRNA decay lowers plasma lipids through silencing of lipid metabolism genes [53].

### 2.2. The Kinase PERK and Transcription Factor ATF4 and CHOP

PERK is a type 1 transmembrane protein and a member of the eukaryotic initiation factor 2α (eIF2α) kinase family that is typically activated through recruitment of chaperone BiP away from PERK, leading to oligomerization and activation of the cytosolic kinase domain [54]. Protein structural analysis also shows that PERK’s luminal domain can recognize and selectively interact with misfolded protein and thereby trigger oligomerization and activation [55,56]. PERK is activated to phosphorylate eIF2α at Ser51 (p-eIF2α), which briefly attenuates initiation of mRNA translation, to reduce the ER protein folding load [29]. Although it reduces global protein synthesis, eIF2α selective upregulates the translation of a subset of mRNAs, the best-studied of which is the transcription factor ATF4 [57]. ATF4 transcriptionally upregulates several UPR-target genes that mediate antioxidant responses and amino acid synthesis and transport [58,59]. ATF4 also upregulates transcription factor CHOP, which forms heterodimers with ATF4 to mediate pro-apoptotic signaling activation [58]. Moreover, ATF4 and CHOP induce transcription of growth arrest and DNA-damage-inducible protein 34 (GADD34) to direct p-eIF2α dephosphorylation and restart global mRNA translation, which is essential if cells are to survive an acute ER stress challenge [60]. 

Germline ablation of PERK leads to 30–40% prenatal mortality and at least 23% postnatal mortality [61]. The surviving newborn Perk^−/−^ mice are of normal size but exhibit severe postnatal growth retardation [61]. Consistently, homozygotes of Atf4^−/−^ pups generally die during the first hour after birth, although excess mortality occurs throughout the first 3 weeks of life [62]. Liver-specific knockout studies show that PERK/ATF4/CHOP signaling also regulates lipid homeostasis [63]. Triglyceride levels are increased in livers of the PERK liver-specific knockout mice through regulating Very Low-Density Lipoprotein Receptor (VLDLR) expression upon tunicamycin treatment [64]. However, ATF4 overexpression induces early onset of hyperlipidemia and hepatic steatosis in zebrafish [65]. Aged female Chop^−/−^ mice reportedly gain much more weight with increased adiposity and hepatic steatosis [66]. CHOP mediates insulin resistance by modulating adipose tissue macrophage polarity [67].

### 2.3. Transcription Factor ATF6α 

ATF6 is a type II ER transmembrane protein, with a cytoplasmic N-terminus that contains a basic leucine zipper (bZIP) domain that functions as a transcription factor following regulated intramembrane proteolysis by site-1 and site-2 proteases (S1P and S2P) in ER-stressed cells [30]. Two Atf6 genes, Atf6α and Atf6β, are expressed ubiquitously, with no obvious phenotype in mice lacking either individual isoform [68,69]. Interestingly, the combined deletion of Atf6α and Atf6β causes a very early embryonic lethality, suggesting these genes provide essential complementary functions in early embryonic development [70]. Transcriptional induction of ER chaperone and ERAD genes such as BiP, glucose-regulated protein 94 (GRP94) and E3 ubiquitin-protein ligase HRD1 is mostly mediated by ATF6α binding to the cis-acting ER Stress Response Element (ERSE) consensus sequence [71,72]. 

Intriguely, pharmacological challenge of ATF6α deficiency mice, but not ATF6β-null mice, with ER stress leading to aggravated hepatosteatosis and death in the liver [73]. In hepatocytes, ATF6α interacts with peroxisome proliferator-activated receptor α (PPARα) to enhance its transcriptional activity, upregulating hepatic fatty acid oxidation [74]. Moreover, ATF6α suppresses SREBP2 transactivation in hepatocytes to downregulate lipogenesis [75]. ATF6α also reduces hepatic glucose output by disrupting the CREB-CRTC2 interaction and thereby inhibiting CRTC2 occupancy over gluconeogenic genes [76]. 

### 2.4. Transcription Factor CREBH

CREBH, encoded by the CREB3L3 gene, is also an ER-residing transcription factor that is robustly expressed in the liver [31]. CREBH contains an ER transmembrane domain, a transcription-activation domain and a basic leucine zipper (bZIP) domain [31,43]. Similar to ATF6, in response to ER stress, CREBH is activated by intramembrane proteolysis and translocates from the ER to the Golgi apparatus, where it is cleaved by the S1P and S2P, releasing bZIP domain to induce transcriptional activity [31,77].

CREBH is a transcription factor that is key in the regulation of hepatic lipid accumulation [78,79]. Hepatic CREBH bZIP domain overexpression significantly decreases the TG levels in plasma and increases hepatic TG levels compared to WT controls [80]. Consistently, the levels of plasma TG in the CREBH null mice are dramatically increased compared with those of the WT mice [80]. Under nutrient starvation, a condition that stimulates lipolysis, the levels of ketone, 3-hydroxybutyric acid, a product of FA oxidation in the plasma, are slightly reduced in the CREBH-null mice, compared with control mice, suggesting that CREBH deletion leads to a defect in TG lipolysis, resulting in higher levels of plasma TG [80]. CREBH regulates hepatic lipid accumulation by directly modulating key metabolic enzyme expression [81]. CREBH directly regulates the transcriptional activation of apolipoprotein, APOC2, APOA4, APOA5 and cell death activator CIDE-3/FSP27 by binding the CRE binding motifs in their gene promoters [81,82]. Moreover, CREBH regulates hepatokine FGF21 expression and subsequently regulates glucose and lipid metabolism [43,83]. Lysine 294 ubiquitination and acetylation of CREBH regulate its stability and transcription activity [43,84].

## 3. Pathogenesis of Hepatocellular Carcinoma

Hepatocellular carcinoma (HCC) is a primary malignancy of the liver and the third leading cause of cancer mortality worldwide [85,86]. HCC is typically diagnosed at advanced stages, with a median survival following diagnosis of approximately 6 to 20 months [85]. HCC is now the third leading cause of cancer deaths worldwide, with over 500,000 people affected [86]. The incidence of HCC is highest in Asia and Africa, where the endemic high prevalence of hepatitis B and hepatitis C strongly predisposes to the development of chronic liver disease and subsequent development of HCC [86,87]. In the United States, the current 5-year survival of HCC patients is only 10% [88]. More importantly, despite the advances in the treatment of many other types of tumors, the advanced HCC five-year survival rate has not been significantly improved during the last 30 years [89]. Major risk factors for HCC include chronic HBV (hepatitis B virus) and HCV (hepatitis C virus) infections, chronic alcohol consumption and non-alcoholic fatty liver disease (NAFLD) (Figure 2) (Table 1) [85,86]. Other risk factors for the development of HCC are alpha1-antitrypsin deficiency, Wilson’s disease, hereditary hemochromatosis, primary biliary cirrhosis and autoimmune hepatitis (Figure 2) [90]. The epidemiologic distribution of these risk factors varies according to geographic location and host-specific factors [90,91]. 

### 3.1. Viral Hepatitis

Chronic hepatitis B virus (HBV) and hepatitis C virus (HCV) infections are the most important causes of HCC, accounting for about 80% of cases worldwide [92]. Chronic HBV infection accounts for approximately 80% of virus-associated HCC cases and virtually all of childhood HCC, especially in Africa and East Asia, while HCV infection, involved in about 20% of the total HCC cases, seems to be mainly related to HCC development in Western Europe and North America [91]. Prospective cohort studies have shown a significant increase in the risk of developing HCC among persons chronically infected with HBV or HCV [91,93]. Between 70 and 90% of the HBV-infected individuals who develop HCC have cirrhosis secondary to chronic necroinflammation [91,93]. HCV also increases the risk of HCC by inducing fibrosis and, eventually, cirrhosis [91]. 

In HBV-infected hepatocytes, large amounts of HBV-expressing proteins are synthesized and folded in ER, leading to ER dysfunction and resulting in ER stress [94]. Clinically, ground-glass hepatocyte (GGH), whose mutant surface proteins (preS1 and preS2 mutants) accumulate in the ER, represents a histological hallmark of chronic hepatitis B virus infection (Table 1) [95]. Many studies report that pre-S deletion has positive relationships with HCC [96]. Pre-S surface antigen variant induces PERK-ATF4-CHOP signal activation, ER-derived oxidative stress and apoptosis [97]. Moreover, transit or stable expression of HBx, a regulatory X protein, induces ATF6 and IRE1α-XBP1s pathway activation [98,99]. Moreover, CREBH is activated by HBx binding, leading to a synergistic effect on the expression of AP-1 target genes through c-Jun induction [100]. HBx-mediated activation of these pathways of UPR probably promotes HBV replication and expression in liver cells [99,100]. 

HCV replication also disrupts normal ER functions and modulates ER stress signaling activation (Table 1) [101,102]. XBP1 expression is elevated in cells carrying HCV subgenomic replicons, but XBP1 trans-activating activity is repressed [103]. This prevents the IRE1α -XBP1 transcriptional induction of EDEM (ER degradation-enhancing α-mannosidase-like protein), which is a type II ER transmembrane protein that binds directly to and enhances the degradation of misfolded proteins through the ERAD pathway [103]. Consequently, misfolded proteins are stable in cells expressing HCV replicons [103]. In addition, Hepatitis C virus core protein activates PERK and ATF6 UPR pathway, leading to autophagy induction by increasing MAP1LC3B and ATG12 expression [104]. HCV induces transforming growth factor β1 (TGFβ1) through activation of IRE1α -JNK and CREBH pathway [101,105]. 

### 3.2. Alcoholic Fatty Liver Disease (ALD)

Alcoholic beverages are classified as a human carcinogen and act synergistically with pre-existing underlying chronic liver disease to further increase the risk of HCC [115]. Heavy drinkers (50 g/day) of ethanol exhibit a 1.4-fold risk for HCC, and chronic alcohol use of more than 80 g/day for longer than 10 years increases the risk of HCC by 5-fold [116,117]. Genetic variations in the alcohol metabolizing enzymes, such as alcohol dehydrogenase 1C (ADH1C) and aldehyde dehydrogenase (ALDH2), which modulate the resulting amount of the carcinogenic aldehyde, are thought of as potential inherited markers of HCC [118]. Generation of tissue-damaging reactive oxygen species (ROS) from alcohol oxidation has been identified as an important mechanism in hepatocarcinogenesis [119]. ROS promotes HCC development via damaging cellular macromolecules and forming lipid peroxides [120,121]. It also stimulates the production of cytokines, inflammation, cell proliferation and upregulates angiogenesis [122]. 

Alcohol is mainly metabolized in the liver, and ethanol metabolism directly impairs ER structure and function in hepatocytes [123]. More evidence shows that alcohol consumption causes hepatic ER stress along with steatosis, inflammation and apoptosis [123]. Moreover, reducing ER stress has been shown to decrease alcoholic liver injury [123]. Interestingly, alcohol feeding only induces IRE1α-Xbp1s pathway activation in the pancreas but not in the liver [124]. However, alanine aminotransferase (ALT) level is elevated in XBP1-deficient mice fed ethanol [124]. Hepatic gene expression profiling of ethanol-fed mice demonstrates upregulated mRNA abundance of UPR target genes after 4 weeks of ethanol feeding [125]. CHOP deficient mice have a remarkable absence of hepatocellular apoptosis in response to alcohol feeding but no protection against hyperhomocysteinemia, ER stress and fatty liver [106]. This suggests a role for CHOP-mediated hepatocyte apoptosis in ethanol-induced liver injury. Moreover, PERK-ATF4-induced upregulation of nicotinamide methyltransferase (NNMT) contributes to alcohol-related fatty liver development [107]. In Zebrafish, blocking ATF6 larvae prevents alcohol-induced steatosis, and ATF6 overexpression induces genes that drive lipogenesis, increases lipid production and causes steatosis [108]. CREBH-knockdown mice are protected against alcohol-induced perturbation of bile acid homeostasis [109]. 

### 3.3. Non-Alcoholic Fatty Liver Disease (NAFLD) and Nonalcoholic Steatohepatitis (NASH)

In recent decades, NAFLD has become the most common liver disease in developed countries [126,127,128]. NAFLD ranges from simple steatosis in the absence of excessive alcohol intake to non-alcoholic NASH with or without cirrhosis [126,127,128]. The risk of HCC is higher in NAFLD patients than that observed in the general clinical population [126,129]. Most HCC cases in NAFLD develop in patients with cirrhosis [129]. Among patients with NAFLD cirrhosis, HCC risk ranges from 1.6 to 23.7 per 1000 persons per year based on other demographic characteristics [129].

ER stress and dysregulation of the UPR proteins have been implicated in human NAFLD [129,130]. Moreover, studies from animal models of NAFLD support the important role of the three UPR pathways in the development and progression of NAFLD. Hepatocyte-specific Ire1α^−/−^ mice develop severe hepatic steatosis and insulin resistance under high-fat diets (HFDs) [110]. IRE1α deficiency increases the abundance of a subset of miRNA clusters in the steatotic livers of HFD-fed mice and patients [110]. Hepatocyte Xbp1 deficiency increases liver injury in mice fed with HFDs/high-carbohydrate diets (HCD) [131]. ATF4 liver-specific knockout mice are protected from HFD or HCD-induced liver steatosis [111]. Mechanistically, ATF4 deficiency suppresses HCD-induced SCD1 and HFD induced cytochrome P450, family 2, subfamily, polypeptide 1 (CYP2E1) expression [111,112]. Overexpression of dominant-negative ATF6 increases susceptibility to development hepatic steatosis and insulin resistance in HFD or HCD [74]. After having been fed an HFD, a massive accumulation of hepatic lipid metabolites and significant increases in plasma TG levels are observed in the CREBH knockout mice. Along with the hypertriglyceridemia phenotype, CREBH-deficient mice display significantly reduced body-weight gain, diminished abdominal fat and increased nonalcoholic steatohepatitis activities [80].

UPR also drives key features of progressive NASH including inflammation and cell death under long-term NAFLD development [132]. Upon ER stress, activated IREα activates JNKs and NF-κB, which are implicated in the transcriptional activation of pro-inflammatory pathways [113]. In NASH, NF-κB has been identified as a central link between hepatic injury and fibrosis and even favoring progression to HCC [133]. Genetic ablation of NF-κB regulators in mouse models leads to spontaneous liver injury, fibrosis and HCC [134,135]. Moreover, liver-specific CREBH KO mice display severe hepatitis in MCD diet feeding without an increase in liver lipid contents [136]. 

### 3.4. Alpha1-Antitrypsin Deficiency (AATD)

Alpha-1-antitrypsin (A1AT) is a protein mainly produced in the liver that protects the lungs from damage caused by activated enzymes [137]. AATD is the most common genetic cause of metabolic liver disease in children and the most frequent inherited indication for liver failure and transplantation in the pediatric population [138]. However, only approximately 10% of homozygosity for AATD, usually of the genotype PiZZ, have marginal deviations in liver test results [139,140]. Generally, AATD adults show clinical manifestations of chronic liver disease during middle or old age [140]. Moreover, heterozygous carriage of the AATD PiMZ variant increases the risk to develop liver cirrhosis [141]. Autopsy studies conducted on PiZZ patients show that over one-third of elderly patients have developed cirrhosis and primary liver cancer [142,143]. Furthermore, AATD is a risk factor for HBV/HCV infection that promotes advanced liver disease and consequently hepatocellular carcinoma [144].

Hereditary deficiency of A1AT is a consequence of the accumulation of polymers of A1AT mutants in the endoplasmic reticulum of hepatocytes [144]. Accumulation of unfolded Z alpha1-antitrypsin inside the endoplasmic reticulum (ER) induces ER proteotoxic stress and increases the incidence of liver injury, fibrosis/cirrhosis and carcinogenesis [145]. Interestingly, although A1AT mutants can accumulate dramatically within the ER and show increased interaction with the chaperone BiP, many fail to constitutively activate the UPR by itself [146]. However, the cells expressing A1AT Z or H334D mutants exhibit hypersensitivity to both pharmacological and physiological ER stressors [147]. Recently, gene array profiling shows that CHOP is upregulated in the liver of PiZ transgenic mice [114,148]. Increased CHOP levels are also detected in diseased livers of children homozygous for the Z allele [114]. CHOP and c-JUN upregulate A1AT expression and play an important role in hepatic disease by increasing the burden of proteotoxic A1AT Z mutants, particularly in the pediatric population [114].

## 4. Targeting ER Stress for HCC Treatment

The ER stress and UPR play important roles in the regulation of cell fate and have become novel signaling targets for HCC treatment. IRE1α in stellate cells is activated by co-culturing with HCC cells [149]. Inhibiting IRE1α-endonuclease activity decreases tumor burden in a mouse model for hepatocellular carcinoma [150]. The expression of XBP1s is increased in HCC cell lines and tissues [151]. Overexpression of XBP1s promotes epithelial–mesenchymal transition (EMT) and metastasis of HCC cells [151]. Moreover, the PERK pathway is robustly activated in HCC [152]. PERK inhibitor (GSK2656157) administration induces ER-stress-mediated cell death and HCC regression [152]. Melatonin increases the sensitivity of HCC to Sorafenib through the PERK-ATF4-Beclin1 pathway [153]. No reported studies have targeted ATF6 and CREBH in HCC treatment, probably because of the lack of availability of specific chemical inhibtors.

## 5. Conclusions

Chronic UPR/ER stress contributes to the pathogenesis of HCC and disease development [149,154]. Multiple risk factors are commonly associated with carcinogenesis of HCC such as HBV/HCV viral infections, excessive alcoholic consumption, obesity and NAFLD [13,14,15,16,17,155]. However, the precise contribution of each of the factors to ER stress induction is not clear, and their importance to HCC may depend on ER stress duration, presence or absence of genetic factors, cross-talks with other pathogenic pathways, and liver disease stages. Emerging evidence indicates a direct involvement of long-term ER stress in liver hepatocellular tumorigenesis [149]. However, the causative or resultant role of ER stress in HCC development and the precise contribution of each individual UPR pathway are incompletely understood. Moreover, hepatocarcinogenesis is a complex multistep process involving multiple different signaling cascades’ activation. Telomere shortening, oncogenic copy number variants, single-nucleotide variants and epigenetic modifications are also well-established molecular mechanisms for HCC tumorigenesis [156]. The molecular crosstalk between these factors and ER stress needs to be further investigated (Figure 3). It is of vital importance to define the cellular and molecular pathogenesis that driven HCC as current treatments for HCC, such as the tyrosine kinase inhibitor sorafenib, are ineffective therapies, as they only extend median survival by a few weeks to months in a subset of patients [157]. Relieving ER stress may serve as a new specific and effective strategy for preventing or treating HCC. 

## Figures and Tables

**Figure 1 ijms-22-01799-f001:**
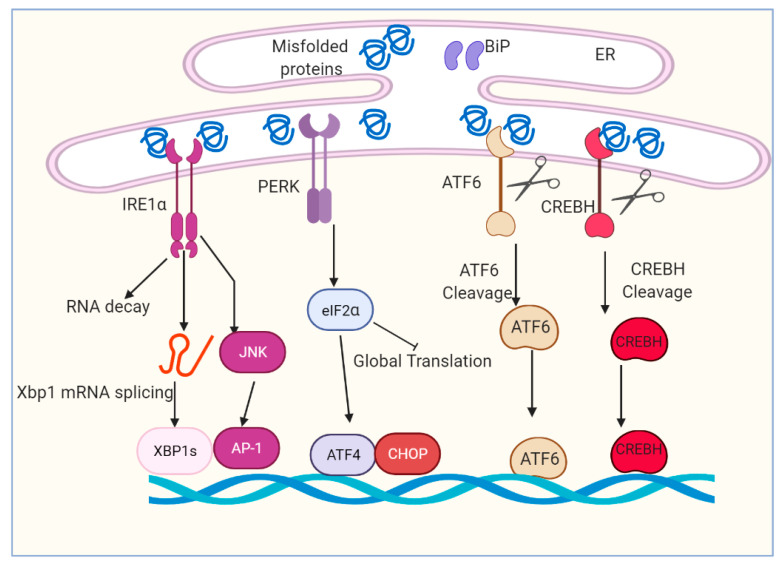
Overview of ER stress and unfolded protein response (UPR) signaling pathway. The accumulation of unfolded and/or misfolded protein induces ER stress and restores ER homeostasis. The cell activates a series of signaling pathways, named the unfolded protein response (UPR). The UPR is regulated by four main proteins: IRE1α, PERK, ATF6 and CREBH. IRE1α, PERK, ATF6 and CREBH are ultimately responsible for the activation of a set of transcription factors, including spliced Xbp1, ATF4, CHOP, cleaved ATF6 and cleaved CREBH.

**Figure 2 ijms-22-01799-f002:**
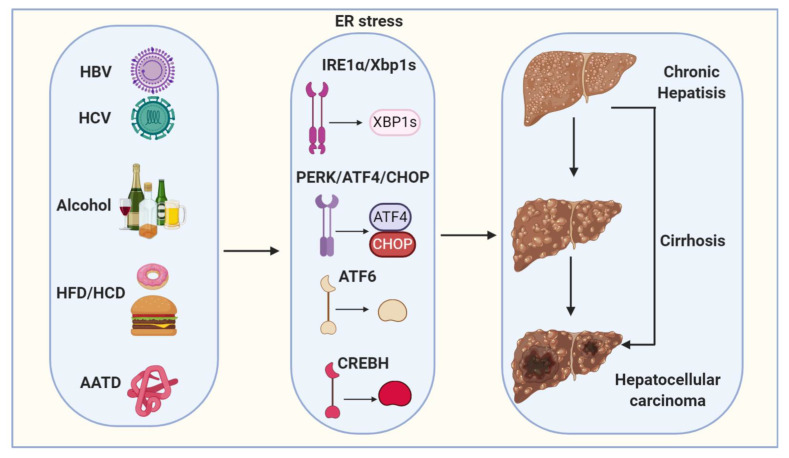
Proposed model depicts ER stress mechanisms linking viral infection, alcohol, high-fat diet/high-carbohydrate diet (HFD/HCD) and Alpha1-antitrypsin deficiency (AATD) with hepatocellular carcinogenesis (HCC). Viral (HBV and HCV) infection, alcohol, HFD/HCD and AATD induce and sustain ER stress. ER stress and steatosis increase reactive oxygen species (ROS) production to cause oxidative stress and inflammation and the subsequent genomic instability, cirrhosis and HCC.

**Figure 3 ijms-22-01799-f003:**
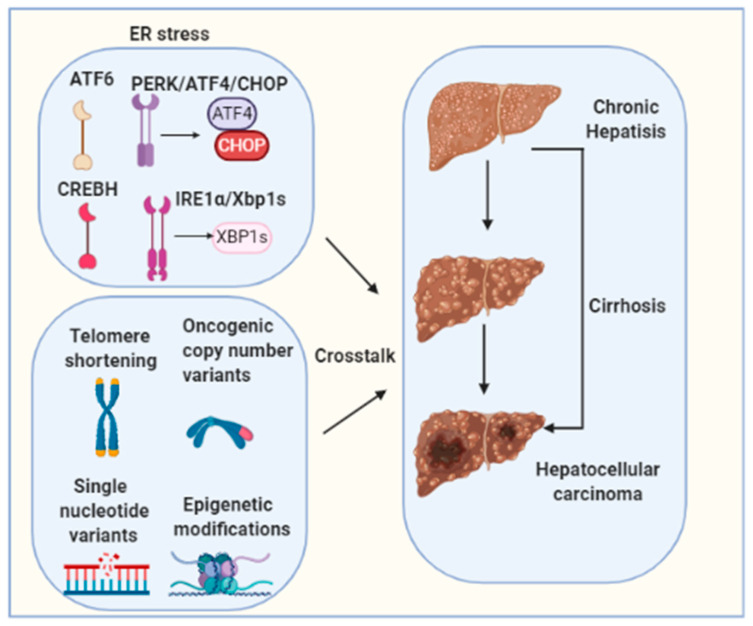
Hepatocarcinogenesis is a complex multistep process involving multiple different signaling cascades’ activation. Telomere shortening, oncogenic copy number variants, single nucleotide variants and epigenetic modifications are well-established molecular mechanisms for HCC tumorigenesis. The molecular crosstalk between these factors and ER stress needs to be further investigated.

**Table 1 ijms-22-01799-t001:** ER stress in the pathogenesis of HCC.

Risk Factor	UPR Pathway Activation	Cellular and Molecular Mechanisms of Liver Injury	Reference
Hepatitis B virus	PERK-ATF4	Oxidative stress and apoptosis	[97]
	ATF6 cleavage	Proliferation of hepatocellular carcinoma cells	[98,99]
	IRE1α-XBP1s		
Hepatitis C virus	IRE1α-XBP1s	Misfolded proteins are more stable	[103]
	IRE1α-JNK	TGFβ1 expression and proliferation	[101,105]
	CREBH cleavage		
	PERK pathway	Autophagy induction	[104]
	ATF6 cleavage		
Alcohol	PERK-ATF4-CHOP	Apoptosis	[106]
		NNMT expression	[107]
	ATF6 cleavage	Lipogenesis and steatosis	[108]
	CREBH cleavage	Perturbation of bile acid homeostasis	[109]
fatty liver (NAFLD/NASH)	IRE1α-XBP1s	Hepatic steatosis and insulin resistance	[110]
		miRNA expression and liver injury	
	PERK-ATF4	SCD1 and CYP2E1 expression; liver steatosis	[111,112]
	ATF6 cleavage	Hepatic steatosis and insulin resistance	[74]
	CREBH cleavage	Nonalcoholic steatohepatitis	[80]
	IREα—JNKs and NF-κB	Hepatic injury and fibrosis	[113]
Alpha1-antitrypsin deficiency (AATD)	CHOP	Upregulates A1AT expression and apoptosis	[114]

## Data Availability

Not applicable.

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
