# Peer review of "Endoplasmic Reticulum Stress Signaling and the Pathogenesis of Hepatocarcinoma"

_ijms, 2021, doi:10.3390/ijms22041799_

Round 1
Reviewer 1 Report
The submitted review is an interesting summary of current knowledge about the role of ER stress in HCC pathogenesis. The paper is generally well written, is quite condense as reader-friendly. I only recommend considering supporting the Conclusions section with summarizing figure?
Author Response
Comment: The submitted review is an interesting summary of current knowledge about the role of ER stress in HCC pathogenesis. The paper is generally well written, is quite condense as reader-friendly. I only recommend considering supporting the Conclusions section with summarizing figure?
Answer: Thanks for the reviewer's comment. The Figure 3 was added to the Conclusion section with summarizing Figure.
Reviewer 2 Report
The review article describes the role of ER stress signaling on the pathogenesis of hepatocarcinoma. The article is interesting, and it addresses a current topic, however, there are a few areas that need to be added/improved before publishing.
- Authors should provide selection criteria of included studies.
- The authors should make a table to summarize the recent studies on ER-stress and HCC.
- Add details of clinical trials on HCC which targets ER-stress/UPR.
- In figure 1, ATF6 and CREBH activated which factors. Please add.
Author Response
We thank for comments from the reviewer and point to point responses are showed below:
Q1: Authors should provide selection criteria of included studies.
Answer1: For a review article, we believe that this is an irrelevant question. Nevertheless, this review has included most if not all recent publications regarding to ER-stress and HCC in experimental animal models and in human.
Q2: The authors should make a table to summarize the recent studies on ER-stress and HCC.
Answer2: The Table was added in the updated Manuscript.
Q3: Add details of clinical trials on HCC which targets ER-stress/UPR.
Answer3: A new section was added to summarize the research which targeting ER stress for HCC treatment.
Q4: In figure 1, ATF6 and CREBH activated which factors. Please add.
Answer4: We thank for reviewer's comment. Figure 1 was revised and cleaved ATF6 and CREBH was added to figure 1.
Round 2
Reviewer 2 Report
The manuscript has been improved in the current revision and the authors addressed my previous concerns. Fig 1 in the revised version is the same as of previous versions. Please add revised fig 1.